# Prognostic Role of the Naples Score in Extensive-Stage Small Cell Lung Cancer: A Missed Opportunity in Inflammation-Based Stratification

**DOI:** 10.3390/jcm14165892

**Published:** 2025-08-21

**Authors:** Fahri Akgül, İvo Gökmen, İsmail Bayrakçı, Didem Divriklioğlu, Aysun Fatma Akkuş, Gizem Bakır Kahveci, Tayyip İlker Aydın, Bülent Erdoğan

**Affiliations:** 1Division of Internal Medicine, Department of Medical Oncology, Faculty of Medicine, Trakya University, Edirne 17000, Turkey; fahriakgul@trakya.edu.tr (F.A.); dr.ismaiilbayrakci@hotmail.com (İ.B.); dr_didemeroglu@hotmail.com (D.D.); aysunfatmadogan@gmail.com (A.F.A.); bakirkahvecigizem@gmail.com (G.B.K.); ilker6125@gmail.com (T.İ.A.); berdoga@hotmail.com (B.E.); 2Department of Medical Oncology, Mehmet Akif Ersoy State Hospital, Çanakkale 17000, Turkey

**Keywords:** Naples Prognostic Score, small cell lung cancer, overall survival, inflammation, prognostic index

## Abstract

**Background:** The Naples Prognostic Score (NPS) is a composite inflammation–nutrition index whose prognostic value has been scarcely examined in extensive-stage small cell lung cancer (ES-SCLC). This study aimed to evaluate the prognostic significance of the NPS in this setting. **Methods:** A retrospective analysis was performed on 142 patients diagnosed with ES-SCLC between March 2014 and June 2024. The NPS was calculated using the neutrophil-to-lymphocyte ratio (NLR), lymphocyte-to-monocyte ratio (LMR), serum albumin, and total cholesterol levels. Patients were classified into three NPS categories (0, 1–2, and 3–4), and subsequently dichotomized into low-risk (0–2) and high-risk (3–4) groups. Survival outcomes were assessed using Kaplan–Meier estimates and multivariate Cox regression models. **Results:** Median overall survival (OS) was significantly longer in the low-risk group compared to the high-risk group (10.3 vs. 6.3 months; *p* = 0.012). High NPS remained an independent predictor of reduced OS (HR: 1.45; 95% CI: 1.02–2.06; *p* = 0.041). The prognostic strength of the NPS was primarily driven by low LMR and hypoalbuminemia, which were individually associated with worse outcomes. **Conclusions:** The NPS may serve as a simple, accessible, and independent prognostic tool in ES-SCLC, potentially aiding in clinical risk stratification and treatment planning.

## 1. Introduction

Small cell lung cancer (SCLC) is a highly aggressive malignancy that, despite accounting for only 13–15% of all lung cancers, contributes disproportionately to global cancer-related mortality [1]. Characterized by rapid tumor proliferation, early metastasis, and high relapse rates, SCLC results in approximately 200,000 deaths annually worldwide [2]. At diagnosis, nearly 70% of patients present with extensive-stage disease (ES-SCLC), where therapeutic options remain limited and the five-year overall survival (OS) rate is estimated to be less than 5%, rarely exceeding 12% even in selected subgroups [3].

Although the incorporation of immunotherapy into systemic regimens has yielded modest survival gains, the prognosis for patients with ES-SCLC remains dismal [4,5]. In this context, accurate and practical prognostic stratification is essential—not only to guide individualized treatment decisions but also to inform clinical trial design and optimize patient selection [6]. Traditional clinical factors such as age, sex, performance status, and metastatic burden offer limited predictive precision, particularly in a disease as biologically heterogeneous as SCLC [7].

Recent attention has turned toward hematologic and biochemical biomarkers related to systemic inflammation and nutritional status [8]. Indices such as the neutrophil-to-lymphocyte ratio (NLR), lymphocyte-to-monocyte ratio (LMR), serum albumin, and total cholesterol have each demonstrated prognostic relevance in various solid tumors [9,10,11]. However, these markers in isolation often lack the robustness needed to inform clinical decisions effectively [12].

The Naples Prognostic Score (NPS), introduced by Galizia et al. in 2017 for gastrointestinal malignancies, combines these four parameters into a single composite index [13]. Its prognostic utility has since been validated across multiple solid tumors, including gastric, pancreatic, colorectal, esophageal, and hepatobiliary cancers [14,15,16]. Owing to its simplicity, accessibility, and reliance on routine laboratory values, the NPS represents a promising tool for real-world clinical application [17].

Considering the aggressive nature and narrow treatment window of ES-SCLC, user-friendly prognostic models like the NPS may hold particular clinical relevance in this population. Previous studies evaluating NPS in SCLC have predominantly included mixed-stage cohorts, combining both limited-stage and extensive-stage disease [18,19]. Given the biological and therapeutic differences between stages, such designs limit the ability to draw stage-specific conclusions. Despite this growing evidence, data regarding the prognostic role of NPS in a homogeneous cohort of patients with ES-SCLC remain scarce, and to our knowledge, no prior study has systematically evaluated NPS exclusively in this group. Therefore, this study aimed to evaluate the prognostic utility of the NPS in patients with ES-SCLC, with the objective of determining its value as a clinically applicable tool for risk stratification in this challenging disease setting.

## 2. Materials and Methods

### 2.1. Study Design and Patient Selection

This retrospective cohort study included patients with histologically confirmed ES-SCLC who received first-line platinum–etoposide chemotherapy at the Department of Medical Oncology, Trakya University, between March 2014 and June 2024. Eligible patients had either de novo metastatic disease at diagnosis or progression to the extensive stage after prior limited-stage treatment, provided they had complete baseline laboratory data required for NPS calculation, complete clinical records, and adequate follow-up.

Patients were excluded if they had received alternative first-line regimens, including platinum–irinotecan in the setting of platinum-sensitive relapse or single-agent chemotherapy in the setting of platinum-resistant relapse, as well as those with concurrent active malignancies, uncontrolled infections, hematologic disorders, or significant hepatic or renal dysfunction. Of the 268 patients initially screened, 81 were excluded due to missing baseline laboratory data or receipt of ineligible chemotherapy regimens. An additional 27 patients were excluded for insufficient follow-up, 7 for comorbidities that could influence inflammatory markers, and 11 for incomplete demographic or clinical information.

Ultimately, 142 patients met all inclusion criteria and were included in the final analysis. Notably, immune checkpoint inhibitors were not reimbursed under the national health insurance system in Turkey during the study period; therefore, none of the patients had received immunotherapy, ensuring treatment homogeneity.

### 2.2. Data Collection and Laboratory Measurements

Demographic, clinical, and laboratory data were extracted from electronic health records, archived patient files, and institutional databases. Baseline laboratory values were defined as the most recent measurements obtained within seven days prior to the initiation of first-line therapy. Hematologic indices were analyzed using the Sysmex XN-1000™ (Sysmex Corporation, Kobe, Japan) automated analyzer, and serum albumin and total cholesterol levels were assessed via enzymatic colorimetric assays (Roche Cobas 8000™, Roche Diagnostics GmbH, Mannheim, Germany). All analyses were performed in accordance with institutional quality control protocols and within standard laboratory reference ranges.

### 2.3. Naples Prognostic Score Assessment

The NPS was calculated based on four parameters reflecting systemic inflammation and nutritional status, using the exact cutoff values defined in the original description by Galizia et al. [13], rather than recalculated within this cohort. One point was assigned for each of the following: an NLR greater than 2.96, an LMR less than 4.44, serum albumin below 40 g/L, and total cholesterol less than or equal to 180 mg/dL. Patients were initially categorized into three groups according to their total NPS score: Group 0 (NPS = 0), Group 1 (NPS = 1–2), and Group 2 (NPS = 3–4).

Although the original three-tier classification yielded a statistically significant global log-rank test (*p* = 0.002) with adequate post-hoc power (~82%), the NPS = 0 subgroup comprised only 10 patients. This small sample size is insufficient for reliable adjustment in multivariable models and increases the risk of unstable estimates and wide confidence intervals. Therefore, despite the lower power (~73%) and a higher *p*-value observed with the binary classification, this approach was adopted for the primary statistical analyses to achieve more balanced groups, reduce sparse data bias, and ensure stable multivariable modeling. In this binary system, patients with NPS scores of 0–2 were classified as the low-risk cohort, while those with scores of 3–4 were considered high-risk. This dichotomous stratification was used in all core analyses, whereas the original three-tier structure was retained for exploratory purposes to assess potential survival gradients.

### 2.4. Outcome Definition

The primary outcome was OS, defined as the time from histological diagnosis to death from any cause. As of the data cutoff date (30 June 2024), all patients in the cohort had deceased, allowing for uncensored survival analysis.

### 2.5. Statistical Analysis

All statistical analyses were conducted using IBM SPSS Statistics software (version 26.0; IBM Corp., Armonk, NY, USA) and MedCalc Statistical Software (version 20.1; MedCalc Software Ltd., Ostend, Belgium). Continuous variables were summarized as medians and interquartile ranges (IQRs), while categorical variables were reported as frequencies and percentages.

Comparisons between the low-risk (NPS 0–2) and high-risk (NPS 3–4) groups were performed using the Mann–Whitney U test for continuous variables, and either the Chi-square or Fisher’s exact test for categorical variables, depending on distribution characteristics.

Overall survival was estimated using the Kaplan–Meier method, and differences between groups were assessed via the log-rank test. Univariate Cox proportional hazards regression models were used to identify potential prognostic variables. Multivariate models were built using backward stepwise elimination (removal threshold: *p* > 0.10), and hazard ratios (HRs) with 95% confidence intervals (CIs) were reported. Two separate multivariate models were developed: Model 1 included the binary NPS classification (low-risk vs. high-risk), while Model 2 included the individual NPS components (NLR, LMR, serum albumin, and total cholesterol). The number of metastatic sites was excluded due to collinearity with organ-specific metastases and conceptual redundancy.

Model discrimination was assessed using Harrell’s concordance index (C-index). The proportional hazards assumption was tested using Schoenfeld residuals and found to be satisfied for all final models.

## 3. Results

### 3.1. Baseline Characteristics

A total of 142 patients with ES-SCLC were included in the analysis (mean age: 60.4 ± 7.9 years), of whom 33.1% were aged ≥65 years. The cohort was predominantly male (92.3%) and composed largely of active smokers (76.1%). At diagnosis, 64.1% of patients had an ECOG performance status (ECOG PS) of 0, and 72.5% had radiologically confirmed extensive-stage disease. Comorbidities were documented in 49.3% of the cohort. Patients received a mean of 5.1 chemotherapy cycles, with 46.5% completing all six planned cycles. Treatment responses were partial in 46.5%, progressive in 30.3%, stable in 18.3%, and complete in 4.9% of cases. Prophylactic cranial irradiation (PCI) was administered to 24.6% of patients.

The most common site of metastasis was bone (50.0%), followed by brain (28.2%), liver (26.1%), adrenal glands (23.2%), and the contralateral lung (17.6%). The mean number of metastatic sites per patient was 2.2, and approximately one-third of patients exhibited single-organ dissemination.

With respect to NPS components, elevated NLR and hypoalbuminemia were present in 62.7% of patients, low LMR in 79.6%, and low total cholesterol in 20.4%. Based on the original scoring system, 7.0% of patients were classified as Group 0 (NPS = 0), 49.3% as Group 1 (NPS 1–2), and 43.7% as Group 2 (NPS 3–4). For analytical purposes, Groups 0 and 1 were merged to constitute the low-risk category, while Group 2 represented the high-risk category.

### 3.2. Comparison Between Risk Groups

Patients were stratified into low-risk (NPS 0–2; *n* = 80) and high-risk (NPS 3–4; *n* = 62) groups. The mean age (60.0 ± 7.5 vs. 60.9 ± 8.4 years; *p* = 0.526) and the proportion of patients aged ≥65 years were similar between groups. ECOG PS of 0 was significantly more common in the low-risk group (72.5% vs. 53.2%; *p* = 0.018). No significant intergroup differences were observed in sex, smoking history, comorbidity burden, disease stage, or treatment characteristics. Although PCI was more frequently administered in the low-risk cohort (30.0% vs. 17.7%), this difference did not reach statistical significance (*p* = 0.093). Objective response rates did not differ significantly (*p* = 0.157), nor did the number or pattern of metastatic sites (all *p* > 0.05). Detailed comparisons of clinical variables between the two risk groups are presented in Table 1.

### 3.3. Overall Survival Analysis

Kaplan–Meier analysis revealed a statistically significant difference in OS between the risk groups (log-rank χ^2^ = 6.307; *p* = 0.012). The median OS was 10.3 months (95% CI: 7.5–13.0) in the low-risk group and 6.3 months (95% CI: 4.6–8.0) in the high-risk group.

All patients had died by the end of the follow-up period, eliminating censoring and allowing for complete survival analysis. The median follow-up time—corresponding to the median OS—was 8.3 months (range: 1.1–40.8; mean: 10.8 ± 8.9). The Kaplan–Meier survival curves illustrating this difference are presented in Figure 1.

### 3.4. Univariate Cox Regression Analysis

Univariate analysis identified several clinical and laboratory parameters significantly associated with reduced OS, including ECOG PS ≥ 1, bone metastasis, low serum albumin, and low LMR. Among site-specific metastases, only bone involvement reached statistical significance. The presence of multiple metastatic sites also predicted worse outcomes. In contrast, age, sex, smoking status, comorbidities, PCI administration, and the pattern of metastatic onset did not demonstrate a significant relationship with OS. Lung metastasis and NLR emerged as borderline predictors, with *p*-values close to the significance threshold (*p* = 0.061 and *p* = 0.098, respectively). Patients with a higher modified NPS (3–4) had significantly poorer OS compared to those with lower scores (see Table 2).

### 3.5. Multivariate Cox Regression Analysis

To determine independent predictors of OS, multivariate Cox models were constructed using variables with *p* < 0.10 in univariate analysis.

#### 3.5.1. Model 1—Prognostic Impact of the Composite NPS

In the first model, we assessed the independent prognostic relevance of the NPS while adjusting for clinical covariates including age (≥65 years), ECOG PS, and site-specific metastases (bone, brain, lung). A higher NPS (3–4 vs. 0–2) was the only variable that retained statistical significance (see Table 3, HR = 1.45; 95% CI: 1.02–2.06; *p* = 0.041). None of the other covariates remained independently associated with OS. These findings underscore the robust prognostic value of the composite NPS in patients with ES-SCLC, independent of conventional clinical factors.

#### 3.5.2. Model 2—Prognostic Role of Individual NPS Components

To evaluate the independent prognostic utility of individual NPS components, a second multivariate model excluded the composite score to avoid collinearity. Variables included serum albumin, LMR, NLR, ECOG status, age, and site-specific metastases. In this model, high serum albumin (≥40 g/L) and LMR (≥4.44) were significantly associated with prolonged OS (see Table 4, *p* = 0.033 and 0.034, respectively), while ECOG PS ≥ 1 approached statistical significance. NLR, metastatic patterns, and age were not independently associated with survival. These results suggest that serum albumin and LMR are the most robust individual predictors among the four NPS components in this population. Harrell’s C-index was calculated for both models: 0.69 for Model 1 and 0.63 for Model 2.

### 3.6. Exploratory Analysis Based on the Original NPS Classification

As an exploratory analysis, patients were stratified into three prognostic groups based on the original NPS: Group 0 (score 0), Group 1 (scores 1–2), and Group 2 (scores 3–4). Although Group 0 comprised only 10 patients, a clear stepwise decline in OS was observed across the groups. The median OS was 18.0, 10.2, and 6.3 months for Groups 0, 1, and 2, respectively (log-rank *p* = 0.002). This gradation in survival is illustrated in Figure 2.

In univariate Cox regression analysis, both Group 1 (HR = 2.52; *p* = 0.012) and Group 2 (HR = 3.36; *p* = 0.001) were significantly associated with poorer OS compared to Group 0. In the multivariate model, these associations remained significant and were even more pronounced (Group 1: HR = 3.60; Group 2: HR = 4.33). Additionally, ECOG PS ≥ 1 (HR = 1.53; *p* = 0.037) and the presence of lung metastasis (HR = 1.64; *p* = 0.036) emerged as independent predictors of worse outcomes. Age, bone metastases, and brain metastases did not retain statistical significance.

These findings suggest that the original NPS classification, particularly when integrated with clinical variables, provides strong prognostic stratification and may enhance individualized risk assessment in patients with ES-SCLC.

## 4. Discussion

Inflammatory cells within the tumor microenvironment (TME) are not passive bystanders; they actively influence tumor dynamics through sustained crosstalk with malignant and stromal elements [20]. Among these, neutrophils—once considered mere first-line defenders—have emerged as paradoxical agents. Although they constitute only 1–2% of circulating leukocytes under normal conditions, tumor-induced expansion, often mediated by TGF-β and related cytokines, transforms them into key enablers of progression [21,22].

By upregulating matrix metalloproteinases (MMPs), neutrophils disrupt extracellular matrix architecture, facilitating both angiogenesis and tissue remodeling. These alterations promote local invasion and systemic dissemination [23]. Similarly, tumor-associated macrophages (TAMs), originating from monocytes, amplify the inflammatory milieu by dampening antitumor immunity and recruiting additional immunosuppressive populations, thereby creating an environment conducive to tumor persistence and therapeutic resistance [24].

Nutritional status, beyond inflammation, exerts a decisive influence on cancer prognosis. Hypoalbuminemia, reflecting both malnutrition and systemic inflammation, has been associated with impaired immune surveillance and elevated oxidative stress [25]. Similarly, reduced serum cholesterol is increasingly recognized as an unfavorable prognostic factor across multiple solid tumors [26]. Such alterations underscore the metabolic toll of cancer and may exacerbate immunologic vulnerability. Hematologic indices like NLR and LMR have gained traction as pragmatic markers of immune imbalance. Elevated NLR indicates heightened neutrophilic inflammation and lymphocyte depletion, whereas reduced LMR signals monocytic dominance—both correlating with poor oncologic trajectories. The NPS synthesizes NLR, LMR, serum albumin, and cholesterol into a unified metric reflecting systemic inflammation and nutritional status. Its integrative design enables superior risk stratification compared to isolated parameters. Its prognostic utility has been validated across gastric, pancreatic, colorectal, gallbladder, and hepatobiliary cancers, supporting its cross-tumor applicability [27,28,29,30].

The NPS demonstrated strong prognostic stratification in our ES-SCLC cohort. Patients classified as low-risk had significantly longer median OS than those in the high-risk group (10.3 vs. 6.3 months; log-rank *p* = 0.012). A clear stepwise decline in survival was also observed using the original three-tier model: median OS was 18.0, 10.2, and 6.3 months for scores 0, 1–2, and 3–4, respectively (log-rank *p* = 0.002). These results confirm the clinical relevance of NPS in advanced-stage disease and support its use as a simple yet powerful tool for initial risk stratification in SCLC.

Our results align with previous studies supporting NPS as an independent prognostic factor in SCLC. For instance, in a retrospective study by Chen et al. involving 128 patients with SCLC, those with a low NPS score (score = 0) had significantly longer OS (19.8 vs. 8.45 months), and NPS emerged as an independent prognostic factor in multivariate analysis [18]. However, although 65.6% of the cohort had extensive-stage disease, the analysis included both limited- and extensive-stage patients, limiting the ability to isolate the NPS’s prognostic impact in advanced-stage cases. Additionally, limited-stage patients were overrepresented in the low-risk group (e.g., 74.5% in Group 0), introducing potential stage-related bias.

Similarly, Liu et al. [19] evaluated 179 patients with SCLC, including both limited- and extensive-stage cases, and reported that higher NPS scores were significantly associated with shorter OS. Using a three-tier model (scores 0, 1, and 2), median OS declined progressively: 20.1 months in the NPS = 0 group, 12.6 months in the NPS = 1 group, and 8.4 months in the NPS = 2 group [19]. In multivariate analysis, an NPS of 2 was independently associated with inferior survival outcomes (HR = 5.03; *p* < 0.001). However, the low-risk group (NPS = 0) was disproportionately composed of limited-stage cases (48.4%), suggesting a potential stage-related bias.

In contrast, our study offers a more targeted assessment within a homogeneous ES-SCLC population, minimizing stage-related confounding and enabling a more accurate evaluation of NPS as a prognostic tool.

Beyond SCLC, the NPS has shown substantial prognostic utility in non-SCLC (NSCLC) as well. Multiple independent studies have confirmed its predictive value across various stages and treatment settings, suggesting broad applicability in lung cancer. Zou et al. analyzed 165 patients with locally advanced NSCLC undergoing surgery after neoadjuvant therapy [31]. Higher NPS scores were significantly associated with shorter OS and DFS, with hazard ratios exceeding 8.7 and 9.6, respectively. Moreover, NPS outperformed other inflammatory and nutritional indices in predictive accuracy (AUC for OS: 0.704). Ren et al. similarly reported inferior survival among 319 patients with surgically treated NSCLC and elevated NPS scores [32]. In their cohort, low NPS correlated with longer median OS (~35 months), whereas high NPS was linked to poor performance status and hypoalbuminemia, reinforcing its biological relevance. A meta-analysis by Wang et al., synthesizing data from six independent cohorts, further substantiated these findings [33]. High NPS was associated with a threefold increase in mortality risk (HR: 3.21) and nearly a fourfold progression risk (HR: 3.81), with a clear dose-response trend. Sensitivity analyses confirmed the robustness of these associations, supporting NPS as a stable and reproducible prognostic tool in NSCLC.

Peker et al. [34] recently reaffirmed the prognostic significance of the NPS in NSCLC. In a cohort of 250 patients, elevated NPS scores were linked to shorter OS (10.4 vs. 18.2 months), particularly in older and advanced-stage individuals [34]. However, their multivariate analysis incorporated both the composite NPS and its individual elements—serum albumin, NLR, and LMR—raising concerns of multicollinearity. While albumin remained significant, the prognostic contributions of NLR and LMR diminished, likely due +to shared variance within the composite index.

To address this, our study adopted a bifurcated modeling approach: one model assessed the composite NPS, while the other analyzed its components independently. This strategy reduced collinearity and allowed for clearer delineation of each variable’s prognostic value. While our analysis focused solely on ES-SCLC, the cumulative evidence from independent cohorts and meta-analyses suggests that NPS may offer broader prognostic utility across lung cancer subtypes.

In our ES-SCLC cohort, LMR and serum albumin remained significant in multivariate analysis, whereas NLR did not. One plausible explanation is that NLR and LMR share a common lymphocyte denominator, but LMR additionally incorporates the prognostic effect of monocytes, which are closely linked to tumor-associated macrophage activity and immune suppression in SCLC. This dual reflection of adaptive immunity (lymphocytes) and pro-tumor inflammation (monocytes) may confer a stronger and more independent prognostic signal than NLR alone. Furthermore, in a uniformly advanced-stage population, neutrophil counts may display less variability, attenuating the discriminative power of NLR compared to markers that integrate nutritional status or monocyte-driven inflammation.

Incorporating clinical variables remains essential in prognostic modeling. In our cohort, ECOG PS was significantly associated with the NPS; patients with an ECOG PS of 0 were more frequent in the low-risk group (72.5% vs. 53.2%; *p* = 0.018). Although ECOG PS ≥ 1 was linked to poorer OS in univariate analysis, this significance diminished in the multivariate model that included NPS, suggesting overlapping prognostic pathways. Furthermore, comparative modeling showed that the composite NPS had superior prognostic discrimination (C-index: 0.69) compared to its individual components (C-index: 0.63), underscoring the additive value of its integrative design.

A key methodological strength of our study is the use of a homogenous cohort consisting solely of patients with extensive-stage SCLC. This design minimized stage-related confounding and allowed for a focused evaluation of NPS as an independent prognostic marker in a high-risk population. Moreover, by examining the NPS as a standalone variable—without embedding it in nomograms or composite models—we underscored its direct clinical utility.

This study presents several methodological limitations. As a single-center, retrospective analysis, it is inherently susceptible to selection bias and data inaccuracies. Nonetheless, rigorous exclusion criteria—such as omitting patients with active infections, severe comorbidities, or incomplete laboratory profiles—enhanced cohort uniformity. Although the original three-tier classification demonstrated adequate statistical power, the small size of the NPS = 0 subgroup limited its suitability for multivariable adjustment, leading to the use of a binary classification for the primary analyses. Despite this adjustment, survival curves remained clearly stratified, with a stepwise decline across risk levels. Furthermore, all variables included in the analyses were obtained at baseline, and treatment-related factors—such as nutritional support interventions and the number of completed chemotherapy cycles—were not incorporated into the prognostic models; the absence of these variables may have introduced residual confounding. Additionally, radiotherapy delivered in the metastatic setting (such as palliative RT or SBRT) was not systematically recorded, and the very small number of such cases precluded meaningful statistical analysis. Given the known immunomodulatory effects of RT, its potential influence on baseline inflammation- and nutrition-based indices, including NPS, cannot be entirely excluded and should be investigated in future prospective studies. Lastly, due to inconsistencies in follow-up data, progression-free survival (PFS) could not be reliably analyzed. Consequently, the study focused exclusively on OS, limiting insights into treatment-specific response dynamics.

## 5. Conclusions

The NPS demonstrated independent prognostic value in patients with ES-SCLC, outperforming individual laboratory markers in multivariate analysis. Its composite design—capturing both systemic inflammation and nutritional status—was associated with higher predictive accuracy, as reflected by a superior Harrell’s C-index.

Owing to its simplicity and availability, the NPS may serve as a practical tool for baseline risk stratification. Its integration into routine clinical assessment could aid treatment planning and support patient selection in clinical trials. Further prospective studies are needed to confirm its utility and generalizability.

## Figures and Tables

**Figure 1 jcm-14-05892-f001:**
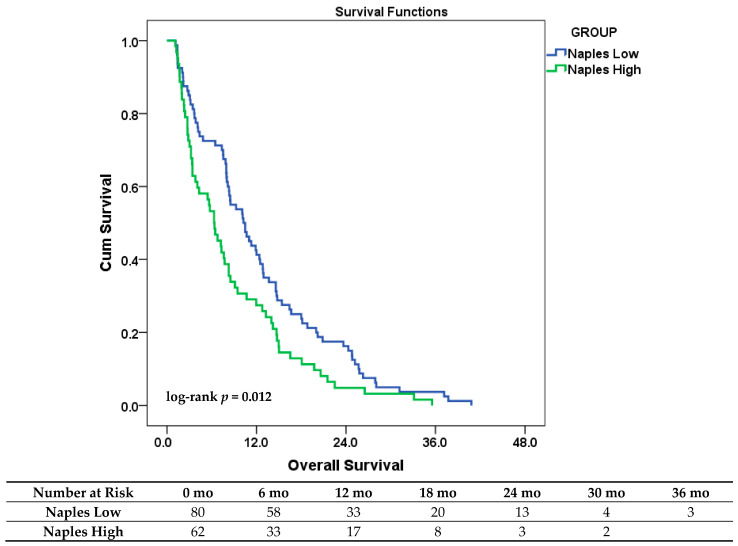
Kaplan–Meier curves for OS in patients with ES-SCLC, stratified by NPS-defined risk groups (low vs. high). **Number at risk**: Counts represent the number of patients still under observation and at risk of the event at each specified time point (in months) for each NPS risk category.

**Figure 2 jcm-14-05892-f002:**
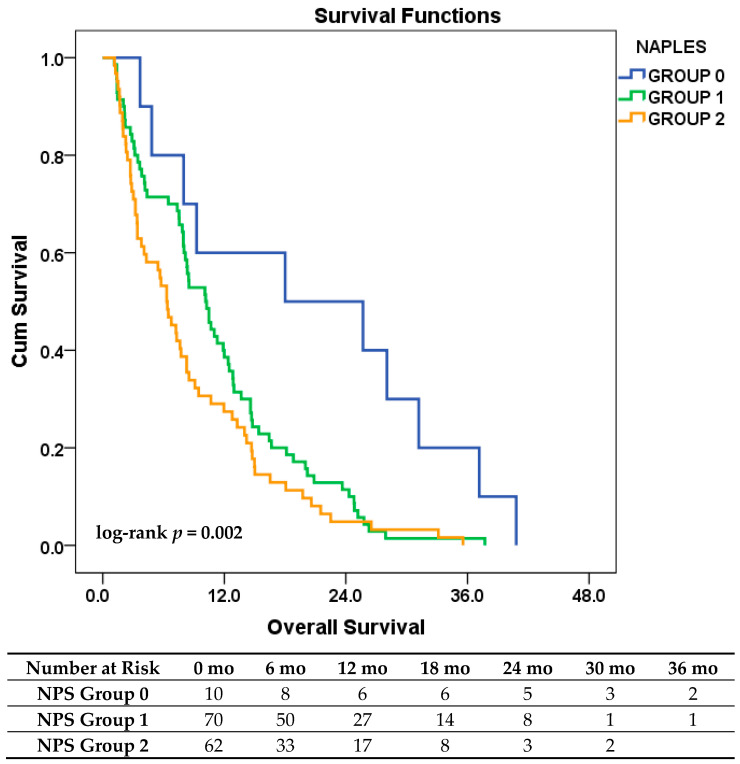
Kaplan–Meier curves for OS in patients with ES-SCLC, stratified by original NPS groups (0 vs. 1–2 vs. 3–4). **Number at risk**: Counts represent the number of patients still under observation and at risk of the event at each specified time point (in months) for each NPS risk group.

**Table 1 jcm-14-05892-t001:** Baseline Demographic and Clinical Characteristics Stratified by NPS Risk Groups.

Clinical Variable	Category	Total (*n*, %)	Low-Risk Group	High-Risk Group	*p*-Value
Age	<65 years	95 (66.9%)	56 (70.0%)	39 (62.9%)	0.472
	≥65 years	47 (33.1%)	24 (30.0%)	23 (37.1%)	
ECOG PS	0	91 (64.1%)	58 (72.5%)	33 (53.2%)	**0.018**
	≥1	51 (35.9%)	22 (27.5%)	29 (46.8%)	
Sex	Female	11 (7.7%)	5 (6.3%)	6 (9.7%)	0.449
	Male	131 (92.3%)	75 (93.8%)	56 (90.3%)	
Smoking Status	Non-current smoker	34 (23.9%)	17 (21.3%)	17 (27.4%)	0.393
	Current smoker	108 (76.1%)	63 (78.8%)	45 (72.6%)	
Comorbidity	Absent	72 (50.7%)	41 (51.2%)	31 (50.0%)	0.883
	Present	70 (49.3%)	39 (48.8%)	31 (50.0%)	
Stage at Diagnosis	Limited disease	39 (27.5%)	23 (28.7%)	16 (25.8%)	0.697
	Extensive disease	103 (72.5%)	57 (71.3%)	46 (74.2%)	
PCI	Not administered	107 (75.4%)	56 (70.0%)	51 (82.3%)	0.093
	Administered	35 (24.6%)	24 (30.0%)	11 (17.7%)	
Number of Metastatic Sites	Single site	45 (31.7%)	25 (31.3%)	20 (32.3%)	0.898
	≥2 sites	97 (68.3%)	55 (68.8%)	42 (67.7%)	
Brain Metastasis	Absent	102 (71.8%)	57 (71.3%)	45 (72.6%)	0.861
	Present	40 (28.2%)	23 (28.7%)	17 (27.4%)	
Bone Metastasis	Absent	71 (50.0%)	44 (55.0%)	27 (43.5%)	0.176
	Present	71 (50.0%)	36 (45.0%)	35 (56.5%)	
Liver Metastasis	Absent	105 (73.9%)	61 (76.3%)	44 (71.0%)	0.477
	Present	37 (26.1%)	19 (23.8%)	18 (29.0%)	
Lung Metastasis	Absent	117 (82.4%)	67 (83.8%)	50 (80.6%)	0.630
	Present	25 (17.6%)	13 (16.3%)	12 (19.4%)	
Adrenal Metastasis	Absent	109 (76.8%)	61 (76.3%)	48 (77.4%)	0.870
	Present	33 (23.2%)	19 (23.8%)	14 (22.6%)	

**Abbreviations**: NPS, Naples Prognostic Score; ECOG PS, Eastern Cooperative Oncology Group Performance Status; PCI, Prophylactic Cranial Irradiation. **Statistical analysis**: *p*-values were calculated using the Pearson chi-square test. A *p*-value < 0.05 was considered statistically significant.

**Table 2 jcm-14-05892-t002:** Univariate Cox Regression Analysis of Overall Survival According to Baseline Clinical and Laboratory Variables.

Clinical Variable	Category	HR (95% CI)	*p*-Value	Reference Category
Age	≥65 years	1.35 (0.94–1.92)	0.101	<65 years
ECOG PS	ECOG ≥ 1	1.43 (1.01–2.03)	**0.047**	ECOG 0
Sex	Male	0.98 (0.53–1.83)	0.957	Female
Smoking Status	Current smoker	1.05 (0.71–1.55)	0.809	Non-current smoker
Comorbidity	Present	1.26 (0.91–1.76)	0.168	Absent
Stage at Diagnosis	De novo	0.93 (0.65–1.34)	0.681	Relapsed
PCI	Yes	0.98 (0.67–1.45)	0.935	No PCI
Number of Metastatic Sites	≥2 sites	1.59 (1.11–2.29)	**0.012**	Single site
Brain Metastasis	Present	1.38 (0.95–2.00)	0.088	Absent
Bone Metastasis	Present	1.41 (1.01–1.98)	**0.044**	Absent
Liver Metastasis	Present	1.08 (0.74–1.58)	0.691	Absent
Lung Metastasis	Present	1.52 (0.98–2.33)	0.061	Absent
Adrenal Metastasis	Present	0.97 (0.65–1.44)	0.868	Absent
NLR	High (≥cutoff)	1.34 (0.95–1.89)	0.098	Low (<cutoff)
LMR	High (≥cutoff)	1.62 (1.06–2.47)	**0.027**	Low (<cutoff)
Serum Cholesterol	High (≥cutoff)	0.86 (0.57–1.30)	0.475	Low (<cutoff)
Serum Albumin	High (≥cutoff)	1.44 (1.02–2.03)	**0.040**	Low (<cutoff)
NPS	High (3–4)	1.54 (1.10–2.15)	**0.013**	Low (0–2)

**Abbreviations**: NPS, Naples Prognostic Score; ECOG PS, Eastern Cooperative Oncology Group Performance Status; PCI, Prophylactic Cranial Irradiation; HR, Hazard Ratio; CI, Confidence Interval; NLR, Neutrophil-to-Lymphocyte Ratio; LMR, Lymphocyte-to-Monocyte Ratio. **Statistical analysis**: Hazard ratios (HRs) and 95% confidence intervals (CIs) were calculated using univariate Cox proportional hazards regression. A *p*-value < 0.05 was considered statistically significant.

**Table 3 jcm-14-05892-t003:** Multivariate Cox Regression Analysis for Overall Survival—Model 1 (Composite NPS Score).

Variable	HR (95% CI)	*p*-Value
High NPS (3–4)	1.45 (1.02–2.06)	**0.041**
Bone metastasis	1.27 (0.88–1.82)	0.203
Brain metastasis	1.23 (0.84–1.78)	0.290
Lung metastasis	0.68 (0.43–1.08)	0.100
ECOG PS ≥ 1	1.19 (0.82–1.73)	0.367
Age ≥ 65	1.20 (0.83–1.73)	0.329

**Abbreviations**: NPS, Naples Prognostic Score; ECOG, Eastern Cooperative Oncology Group; HR, Hazard Ratio; CI, Confidence Interval. **Statistical analysis**: Multivariate Cox proportional hazards regression was performed using the composite NPS classification (low vs. high) along with clinical covariates. Hazard ratios (HRs) and 95% confidence intervals (CIs) are reported. A *p*-value < 0.05 was considered statistically significant.

**Table 4 jcm-14-05892-t004:** Multivariate Cox Regression Analysis for Overall Survival—Model 2 (Individual NPS Components).

Variable	HR (95% CI)	*p*-Value
LMR (High)	1.65 (1.04–2.61)	**0.034**
Serum albumin (High)	1.48 (1.03–2.11)	**0.033**
NLR (High)	1.01 (0.69–1.49)	0.954
Bone metastasis	1.30 (0.89–1.88)	0.164
Brain metastasis	1.18 (0.80–1.73)	0.408
Lung metastasis	0.71 (0.45–1.12)	0.139
ECOG PS ≥ 1	1.43 (0.97–2.11)	**0.068**
Age ≥ 65	1.11 (0.77–1.60)	0.591

**Abbreviations**: NLR, Neutrophil-to-Lymphocyte Ratio; LMR, Lymphocyte-to-Monocyte Ratio; ECOG, Eastern Cooperative Oncology Group; HR, Hazard Ratio; CI, Confidence Interval. **Statistical analysis**: Multivariate Cox regression analysis was conducted using the individual NPS components (NLR, LMR, serum albumin, and cholesterol) along with clinical variables. Hazard ratios (HRs) and 95% confidence intervals (CIs) are shown. Statistical significance was defined as *p* < 0.05.

## Data Availability

Data supporting the findings of this study are available from the corresponding author upon reasonable request.

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
