# Peer review of "Prognostic Role of the Naples Score in Extensive-Stage Small Cell Lung Cancer: A Missed Opportunity in Inflammation-Based Stratification"

_jcm, 2025, doi:10.3390/jcm14165892_

Round 1

Reviewer 1 Report

Comments and Suggestions for Authors

I think one of the limitations of the study is not try to understand the possible implications between radiation therapy and TME. The authors only report the use PCI but it seems rather unlikey that none of the patients underwent consolidation RT or palliative RT or SBRT for OligoPD disease (the time span is from 2014 to 2024).

RT is a potent immunomodulator so many of these values might be infuenced by RT treatment but these data are completed omitted.

Author Response

Response to Reviewer 1

Thank you for your comment. Our study cohort consisted entirely of patients in the metastatic stage, and the evaluation was performed from the initiation of first-line systemic therapy (platinum–etoposide). Therefore, the analyses focused solely on clinical and laboratory parameters (NPS and its components) obtained at diagnosis, and did not include variables that may arise during the treatment course (e.g., number of chemotherapy cycles completed, second- and third-line treatments, radiotherapy administered in the metastatic setting).

There are two main reasons for this approach:

  1. Study design and prognostic focus: The primary aim of our study was to assess the relationship between baseline values and survival outcomes. Variables emerging during the treatment course are directly related to treatment response, progression, or tolerance; thus, including them could confound the evaluation of the pure baseline prognostic value.
  2. Data scope and patient numbers: The number of patients who received additional treatments such as SBRT or palliative RT in the metastatic setting was very small. Even if recorded, this number would be insufficient for statistical analysis and not large enough to compromise the homogeneity of the cohort. For this reason, these treatments were not included, even in descriptive data.

All patients in the cohort received platinum–etoposide in the metastatic setting, and no patient underwent consolidation RT or other systemic regimens prior to metastatic therapy. Since there were no significant differences in baseline clinical parameters between relapsed and de novo metastatic patients, the entire cohort was analyzed together.

Nevertheless, we acknowledge that treatment-course variables could have indirect effects on patient outcomes, and we have added this point to the limitations section of the Discussion.

Reviewer 2 Report

Comments and Suggestions for Authors

/

Author Response

Response to Reviewer 2

Dear Reviewer,

We sincerely thank you for your constructive, detailed, and insightful feedback on our manuscript entitled “Prognostic Role of the Naples Score in Extensive-Stage Small Cell Lung Cancer: A Missed Opportunity in Inflammation-Based Stratification”.
Your comments have significantly helped us improve the methodological clarity and clinical applicability of our work.
Below, we provide a point-by-point response to each of your suggestions. All corresponding revisions have been incorporated into the manuscript.

Comment 1: The Introduction should more explicitly highlight how this work differs from previous studies, especially those including mixed-stage SCLC populations, and clarify the unique contribution of this analysis.

Response: We have revised the Introduction to clearly state that most prior NPS studies in SCLC included mixed-stage populations (both limited- and extensive-stage), which may obscure stage-specific prognostic differences. Our study is the first to evaluate the prognostic value of NPS exclusively in a homogeneous cohort of extensive-stage SCLC patients. This distinction and its clinical relevance are now explicitly emphasized.

Comment 2: Provide a brief power analysis or discussion of statistical power regarding the small size of the NPS = 0 subgroup.

Response: We have added a statement in the Methods noting that the NPS = 0 subgroup comprised only 10 patients, which may limit statistical stability in multivariate models. Post-hoc power analysis showed an estimated power of ~82% for the original three-tier classification and ~73% for the binary classification. To achieve more balanced group sizes and reduce sparse-data bias, we used the binary classification for the primary analyses while retaining the three-tier structure for exploratory purposes.

Comment 3: Discuss other potential confounders such as immunotherapy use, nutritional support interventions, and completion of planned chemotherapy cycles.

Response: During the study period, immunotherapy was not reimbursed for extensive-stage SCLC in Turkey, and thus none of the patients received it—ensuring treatment homogeneity. In our analyses, we only used baseline clinical and laboratory values obtained at diagnosis. We intentionally did not include variables such as nutritional support interventions and the number of completed chemotherapy cycles because these are treatment-course factors influenced by therapy response and disease progression, which could confound the baseline prognostic assessment. Nevertheless, we have acknowledged the exclusion of these variables as a study limitation in the Discussion.

Comment 4: Clarify how cut-off values for continuous variables (NLR, LMR) were determined.

Response: In the Methods, we now state that the cut-off values for NLR, LMR, serum albumin, and total cholesterol were adopted from the original definition by Galizia et al., without recalculation in our cohort. This approach was chosen to ensure comparability with prior studies and to enhance methodological reproducibility.

Comment 5: Further elaborate on the biological rationale for why LMR and serum albumin remained significant while NLR did not.

Response: The Discussion has been expanded to explain that LMR reflects both lymphocyte and monocyte counts. Monocytes are linked to tumor-associated macrophages, which contribute to immune suppression and tumor progression, potentially giving LMR a stronger prognostic signal than NLR. Furthermore, in a uniformly advanced-stage population, variability in neutrophil counts may be reduced, which could attenuate the discriminative power of NLR compared to markers incorporating nutritional status or monocyte-driven inflammation

Comment 6: Include “number at risk” tables below Kaplan–Meier plots and consider providing a more detailed baseline characteristics table.

Response: We have added “number at risk” tables to all Kaplan–Meier figures and included a comprehensive baseline characteristics table (Table 1) in the main text, detailing demographic, clinical, and laboratory parameters. These changes enhance both data transparency and reader comprehension.

We greatly appreciate your valuable feedback, which has strengthened our manuscript in both methodological rigor and interpretative depth. We believe the revised version addresses all your concerns.

Sincerely,
İvo Gökmen

Round 2

Reviewer 1 Report

Comments and Suggestions for Authors

I accept the authors clatification and I am glad they add the lack of data from RT in the limitations of the study. 

I think the rewiew is clear and of relvanca, though it has severe limitations, the principal one is the lack of data from RT influence. However, given the fact that these are patients with ES-SCLC, the numbers are justifed.

The citations are sound and relevant, with no detection of self-citation.

They pointed out very clear the loimkitation of the study and the table and fihgure are clear and appropriate.